# Lactate Protects Intestinal Epithelial Barrier Function from Dextran Sulfate Sodium-Induced Damage by GPR81 Signaling

**DOI:** 10.3390/nu16050582

**Published:** 2024-02-21

**Authors:** Xiaojing Li, Zhijie Yao, Jin Qian, Hongling Li, Haitao Li

**Affiliations:** 1State Key Laboratory of Food Science and Resources, Jiangnan University, Wuxi 214122, China; 7190112018@stu.jiangnan.edu.cn (X.L.); 7210112067@stu.jiangnan.edu.cn (Z.Y.); 6210112061@stu.jiangnan.edu.cn (J.Q.); 2School of Food Science and Technology, Jiangnan University, Wuxi 214122, China; 3College of Food Science and Light Industry, Nanjing Tech University, Nanjing 211816, China

**Keywords:** colitis, GPR81, lactate, intestinal epithelial barrier, MMP9, NF-κB

## Abstract

The dysregulation of the intestinal epithelial barrier significantly contributes to the inflammatory progression of ulcerative colitis. Recent studies have indicated that lactate, produced by gut bacteria or derived from fermented foods, plays a key role in modulating inflammation via G-protein-coupled receptor 81 (GPR81). In this study, we aimed to investigate the potential role of GPR81 in the progression of colitis and to assess the impact of lactate/GPR81 signaling on intestinal epithelial barrier function. Our findings demonstrated a downregulation of GPR81 protein expression in patients with colitis. Functional verification experiments showed that *Gpr81*-deficient mice exhibited more severe damage to the intestinal epithelial barrier and increased susceptibility to DSS-induced colitis, characterized by exacerbated oxidative stress, elevated inflammatory cytokine secretion, and impaired expression of tight-junction proteins. Mechanistically, we found that lactate could suppress TNF-α-induced MMP-9 expression and prevent the disruption of tight-junction proteins by inhibiting NF-κB activation through GPR81 in vitro. Furthermore, our study showed that dietary lactate could preserve intestinal epithelial barrier function against DSS-induced damage in a GPR81-dependent manner in vivo. Collectively, these results underscore the crucial involvement of the lactate/GPR81 signaling pathway in maintaining intestinal epithelial barrier function, providing a potential therapeutic strategy for ulcerative colitis.

## 1. Introduction

Inflammatory bowel disease (IBD) is a chronic inflammation of the gastrointestinal tract, afflicting over 6.8 million people worldwide. It primarily comprises Crohn’s disease, ulcerative colitis (UC), and indeterminate colitis [1]. UC is characterized by diffuse and superficial inflammation in the colon and rectum, with bloody diarrhea being a predominant symptom [2]. The global prevalence of UC was estimated to be 5 million cases in 2023, underscoring its significance as a global public health concern [3]. While common treatments such as 5-aminosalicylate and corticosteroids are frequently employed, clinical observations have highlighted their limited efficacy and notable side effects [4]. Consequently, there is an urgent need to develop novel therapeutic strategies that facilitate the implementation of precision medicine for optimal disease management.

While the exact pathogenesis of UC remains elusive, several factors have been implicated, including genetic predisposition, dysregulated immune response, microbiota, and defects in the gut epithelial barrier [5]. The disruption of the intestinal epithelial barrier is central to the etiology and pathology of colitis [6], making the maintenance of barrier function a promising therapeutic strategy. The integrity of this barrier is primarily upheld by intercellular junctions, such as tight junctions (TJ) and adherens junctions (AJ), along with mucus production [4]. Restoring the expression of TJ proteins, including Claudin-1, zonula occludens-1 (ZO-1), and Occludin, is crucial for treating colitis, as it aids in maintaining barrier function and regulating intestinal permeability [7].Moreover, the extracellular matrix provides structural and biochemical support to adjacent cells, influences the expression of TJ proteins in the epithelium, and plays a role in preserving barrier function [8,9]. Collagen, a principal component of the extracellular matrix, enhances to the structural stability of the intestinal mucosa, thereby influencing its permeability and protective attributes [10]. Matrix metalloproteinases (MMPs), especially MMP9, can compromise the intestinal epithelial barrier by degrading collagen and TJ proteins, facilitating the invasion of inflammatory cells and impeding wound healing [10,11,12]. In the intestinal tissues of IBD patients, MMP9 is significantly elevated, suggesting its potential as an indicator of disease activity. Experimental models of colitis have shown that the targeted deletion or pharmacological inhibition of MMP9 can mitigate colonic damage, indicating that MMP9 may serve as a viable therapeutic target for UC [13].

The interaction between the gut microbiota and intestinal epithelial cells (IECs) is implicated as a significant factor in maintaining intestinal homeostasis and barrier integrity. This interaction involves metabolites such as secondary bile acids, tryptophan metabolites, and short-chain fatty acids (SCFAs) [14]. Lactate, which is also produced by lactic acid bacteria in the gut, is a major component of fermented foods, including pickled Chinese cabbage and yogurt. However, the role of lactate as a metabolic mediator for the intestinal microbiota in protecting the intestinal barrier remains unclear. Beyond its critical role in anaerobic glycolysis, lactate functions both as a substrate to support cell growth and as a signaling molecule that regulates immune responses and cellular homeostasis [15,16,17]. Recent studies have shown that lactate acts as a signaling molecule through G-protein-coupled receptor 81 (GPR81) [16], which is expressed in various cells and tissues, suggesting its likely diverse physiological roles within the body [18]. Researchers have revealed that lactate/GPR81 signaling plays a pivotal role in modulating inflammation in conditions such as acute pancreatitis, acute hepatitis, and colitis and can help prevent histopathological damage [19,20,21,22]. Moreover, GPR81 has been shown to suppress intestinal inflammation by promoting Treg responses and limiting pathological Th1/Th17 responses [21]. It has also been observed that lactate enhances colonic antioxidant capacity and increases butyrate levels, thereby alleviating DSS-induced colitis injury by upregulating the expression of GPR81 [22]. Given the protective effect of intestinal Lactobacilli on the intestinal barrier [23,24], it is speculated that lactic acid bacteria may exert their beneficial effects through their metabolite, lactate.

Therefore, we aim to investigate the role of GPR81 in the intestinal epithelial barrier and further elucidate whether dietary lactate can protect against intestinal mucosal damage by activating GPR81.

## 2. Materials and Methods

### 2.1. Chemicals and Reagents

The Human colon disease tissue microarray (BC05002b) was obtained from Biomax (Rockville, MD, USA). DSS (9011-18-1) was acquired from MP Biologicals (Irvine, CA, USA). The primary antibodies anti-GPR81 (SAB1300089), fluorescein isothiocyanate (FITC)–dextran (68059), tumor necrosis factor-alpha (TNF-α) (H8916), and lactate (71720) were purchased from Sigma-Aldrich (St. Louis, MO, USA). Interleukin-6 (IL-6) (DY406) and TNF-α (DY410) ElISA kits were obtained from R&D Systems (Minneapolis, MN, USA). The myeloperoxidase (MPO) assay kit (A044-1-1), malondialdehyde (MDA) assay kit (A003-1-2), and lipid peroxidation (LPO) assay kit (A106-1-2) were sourced from Nanjing Jiancheng Bioengineering Institute (Nanjing, China). Primary antibodies against MMP9 (ab283575), ERK1/2 (ab184699), Claudin-1 (ab211737), ZO-1 (ab96587), and Occludin (ab216327) were prepared by Abcam (Cambridge, UK). Primary antibodies against IκBα (4814), NF-κB p65 (8242), p-NF-κB p65 (Ser536) (3033), p38 (9212), p-p38 (Thr180/Tyr182) (4511), JNK (9252), p-JNK (Thr183/Tyr185) (4668), and p-ERK1/2 (Thr202/Tyr204) (4370) were purchased from Cell Signaling Technology (Beverly, MA, USA). HRP-conjugated Affinipure Goat Anti-Rabbit IgG(H+L) was purchased from Proteintech (Wuhan, China).

### 2.2. Animal Experiments

With a C57BL/6 genetic background, littermate wild-type (WT) and *Gpr81* whole-body knockout (*Gpr81^−/−^*) mice were purchased from Beijing Vitalstar Biotechnology Co., Ltd. (Beijing, China). Genotypes were verified by PCR agarose gel electrophoresis. The Institutional Animal Ethics Committee of Jiangnan University approved all animal experimental procedures (protocol numbers JN.No20210430c0520531[099], JN.No20220330c0400928[099]).

The DSS studies were conducted with slight adjustments to previously published procedures. Given the increased susceptibility of male mice to experimental colitis compared to female mice, male C57BL/6 mice, aged 6–8 weeks and in good health, were randomly selected for the study. The mice were administered 3% (*w*/*v*) DSS in drinking water for 5 days [25]. For lactate treatment, mice were supplemented with either 250 mg/kg or 500 mg/kg of lactate via gavage alongside the DSS treatment.

### 2.3. Histopathology, Immunohistochemistry, and Immunofluorescence

The colon tissues were preserved in 10% formalin and subsequently embedded in paraffin. Tissue sections were then stained using hematoxylin and eosin (H&E) or Masson’s trichrome for histological examination [26,27]. The slides were immunohistochemically stained with anti-GPR81 (1:100), anti-Claudin-1 (1:200), anti-ZO-1 (1:200), and anti-Occludin (1:200) [28]. Immunofluorescence staining was conducted following the established protocol, and the sections were incubated with anti-MMP9 (1:50) [29].

### 2.4. Epithelial Permeability Assay

The mice underwent a 4 h period of fasting and water deprivation, followed by the administration of 0.6 mg/g body weight of 4 kDa FITC–dextran. After 3 h, serum samples were evaluated for FITC fluorescence levels using excitation at 485 nm and emission at 520 nm [30].

### 2.5. Measurement of Cytokine Levels and Oxidative Stress Markers

Colon tissue was homogenized and centrifuged to collect supernatant. The cytokine content in colonic tissue was assessed using an enzyme-linked immunosorbent assay following the manufacturer’s guidelines. Samples were added to antibody-coated microplates. After incubation and washing, a detection antibody was added, and absorbance was measured using a spectrophotometer for protein quantification. MPO, MDA, and LPO levels were detected using commercial assay kits according to the protocols. Absorbance was read at appropriate wavelengths using a spectrophotometer, and concentrations were calculated based on standard curves.

### 2.6. Cell Culture

IECs from the WT mice and *Gpr81*^−/−^ mice were extracted and cultivated as previously described [31,32]. Colons were exposed to 75 μg/mL of collagenase type XI and 20 μg/mL of neutral protease for 2 h. The resulting pellets were centrifuged after suspension in DMEM-2% sorbitol. Subsequently, the cells were treated with 100 ng/mL of TNF-α for 6 h or as recommended.

### 2.7. Quantitative Real-Time PCR

The UNlQ-10 Column TRIzol Total RNA Isolation Kit (B511321, Sangon Biotech, Shanghai, China) was employed to extract total RNA following the manufacturer’s instructions, and this was subsequently converted into cDNA using the HiScript III All-in-one RT SuperMix (R333-01, Vazyme, Nanjing, China), as previously outlined [33]. The primer sequences utilized for quantitative real-time PCR (qPCR) can be found in Table 1. Data were analyzed using the threshold cycle method (2^−ΔΔCT^), with β-actin serving as the internal reference gene.

### 2.8. Western Blotting

Proteins were extracted from the samples using RIPA lysis buffer supplemented with proteinase and phosphatase inhibitors. Following protein quantification, the samples were denatured and separated by SDS-PAGE, then transferred onto nitrocellulose membranes. Protein bands were visualized using an enhanced chemiluminescence detection system after blocking and incubation with specific primary (1:1000) and secondary antibodies (1:2000) [34].

### 2.9. Statistical Analysis

Data analysis was conducted utilizing GraphPad Prism 8.0. ImageJ 1.54f was used for the quantitative analysis of immunohistochemistry and Masson’s trichrome staining. The results are expressed as the mean ± SEM. Two-group comparisons were assessed using a two-tailed t-test, while comparisons involving three or more groups were evaluated using a one-way ANOVA. Statistical significance was defined as *p* < 0.05 and indicated as * *p* < 0.05, ** *p* < 0.01, and *** *p* < 0.005. “NS” denotes non-significant differences.

## 3. Results

### 3.1. Downregulation of GPR81 Expression in the Intestinal Mucosal Tissue of Colitis Patients and Mouse

To investigate the expression of GPR81 in the intestinal mucosal tissue of colitis patients, we initially analyzed GPR81 mRNA expression using public Gene Expression Omnibus (GEO) databases (GDS3119 and GDS3268). The findings revealed a significant downregulation of GPR81 mRNA levels in the inflamed colon tissue of UC patients compared to non-inflamed tissues (Figure 1A). Additionally, we examined the clinical relevance of GPR81 expression in a human colon tissue microarray of colitis samples. Immunohistochemistry staining demonstrated a weaker expression of GPR81 in the chronic colitis samples than in the normal colonic tissues (Figure 1B,C). Consistent with these results, both qPCR and Western blot analyses indicated a similar decrease in mRNA and protein levels of GPR81 in the mouse colitis tissues compared to the normal tissues (Figure 1D,E). Overall, these findings suggest that GPR81 is downregulated in the intestinal mucosal tissue of colitis patients and mouse, indicating its potential involvement in the development of colitis.

### 3.2. Gpr81-Deficient Mice Exhibit High Susceptibility to DSS-Induced Colitis

The above findings prompted us to utilize genetically engineered mouse models (GEMs) to elucidate the potential functions of GPR81 in colitis. We used *Gpr81* whole-body knockout mice and performed a qPCR analysis on their colon tissue to confirm genotypes (Figure 2A,B). Specifically, we administered 3% DSS in the drinking water of both the *Gpr81*^−/−^ and WT mice for 5 days, followed by a recovery period of 9 days. Intestinal permeability assays and colon collection were conducted on the second day after the cessation of DSS administration (Figure 2C). This chemically induced colitis model is commonly used due to its clinical similarities with human UC [25]. The susceptibility to colitis was assessed daily by monitoring body weight changes and observing clinical features such as diarrhea and occult stool bleeding. Prior to DSS administration, no differences were observed between the WT and *Gpr81*^−/−^ mice. Additionally, there was no significant difference in drinking water consumption between the WT mice and the *Gpr81*^−/−^ mice during the DSS treatment (Figure 2C). However, upon DSS administration, the *Gpr81*^−/−^ mice exhibited significantly greater weight loss compared to the WT mice and showed delayed recovery (Figure 2D). Furthermore, the *Gpr81*^−/−^ mice displayed more frequent instances of diarrhea and rectal bleeding than the WT mice, as indicated by their elevated DAI scores (Figure 2E,F). Intestinal barrier function was assessed using the FITC–dextran assay [35]. In the DSS-treated mice, the *Gpr81*^−/−^ mice showed a FITC–dextran value of 3.89 ± 0.18 μg/mL, whereas the WT group exhibited a value of 1.39 ± 0.16 μg/mL (Figure 2G). The colons of the *Gpr81*^−/−^ mice presented a significant reduction in length compared to those of the WT mice (Figure 2H). The histological staining of colon tissue sections further corroborated the increased severity of colitis in the *Gpr81*^−/−^ mice, showing complete crypt destruction and epithelial ulceration when compared to the WT mice (Figure 2I). A histopathological examination revealed significantly higher scores for inflammation, ulceration, and crypt distortion in the colons of the *Gpr81*^−/−^ mice relative to the WT mice (Figure 2J). These findings underscore the protective role of GPR81 in DSS-induced colitis.

### 3.3. Gpr81 Deficiency Impairs Intestinal Epithelial Barrier Function in Colitis

A compromised intestinal barrier, primarily due to oxidative stress and inflammatory responses within the intestine, disrupts intestinal homeostasis and exacerbates disease progression [36]. The results indicated that the *Gpr81*^−/−^ mice exhibited a significant increase in MPO activity, as well as enhanced production of MDA and LPO (Figure 3A). Furthermore, we evaluated the levels of inflammatory cytokines in the colon tissues. Our data demonstrated that both the mRNA and protein levels of TNF-α and IL-6 in the colon tissues of the *Gpr81*^−/−^ mice were significantly higher than those in the WT mice after the DSS treatment (Figure 3B). To further investigate the protective function of GPR81 against DSS-induced intestinal barrier damage, we assessed the mRNA levels of classic secreted mucin (*Muc2*), TJ proteins (*Claudin1*, *Zo1*, and *Occludin*), and AJ proteins (*E-cadherin* and *β-catenin*). There was no significant difference in the *Muc2* level between the *Gpr81*^−/−^ mice and WT mice, while the levels of β-catenin slightly decreased (Figure 3C). However, the *Gpr81*^−/−^ mice exhibited significantly decreased mRNA levels of TJ proteins (*Claudin1*, *Zo1*, and *Occludin*) compared to the WT group (Figure 3C). The expression of TJ proteins in the colon tissues was further assessed by IHC. The results revealed a lower percentage of brownish-yellow areas corresponding to Claudin-1, ZO-1, and Occludin in the *Gpr81*^−/−^ colitis mice, indicating a reduced expression of TJ proteins (Figure 3D,E). These findings suggest that GPR81 alleviates DSS-induced oxidative stress and inflammatory responses, thereby contributing to the preservation of intestinal epithelial barrier function.

### 3.4. Gpr81 Deficiency Promotes MMP9 Production in Colitis

MMP9 has been identified as a biomarker of intestinal inflammation and is known to exacerbate colitis severity by increasing the permeability of intestinal epithelial TJs [13,37]. Thus, we further explored the expression of MMP9 in intestinal mucosal tissue. The data indicated that the level of *Mmp9* in the *Gpr81^−/−^* mice was nearly tenfold higher compared to the WT group after the DSS treatment (Figure 4A). The immunofluorescence analysis of colons from the *Gpr81*^−/−^ colitis mice also showed a significant accumulation of MMP9 (Figure 4B). MMP9 plays a role in intestinal homeostasis through its involvement in collagen degradation [38]. We observed reduced collagen staining in the *Gpr81^−/−^* mice relative to the WT mice after DSS-induced colitis (Figure 4C,D). Consistent with the findings in Figure 1, the protein levels of GPR81 decreased after the DSS treatment. Furthermore, *Gpr81* knockout resulted in increased protein expression of MMP9 and decreased expressions of Claudin-1, ZO-1, and Occludin in the colitis mice (Figure 4E). These findings suggest that GPR81 may exert a protective role in maintaining intestinal barrier integrity by modulating MMP9 expression.

### 3.5. Lactate/GPR81 Signaling Modulates TNF-α-Induced MMP9 Expression in IECs via the NF-κB Pathway

The transcription factor NF-κB is known to induce the expression of MMP-9 [39]. TNF-α, a quintessential pro-inflammatory cytokine, plays a critical role in the pathogenesis of various inflammatory disorders, including IBD [40]. By treating healthy IECs with TNF-α, we aim to mimic an inflammatory stimulus in vitro that closely resembles the inflammatory milieu observed in vivo during active IBD. Consistent with previous studies [39,41], our observations confirmed that TNF-α treatment led to the degradation of IκBα in IECs, culminating in an enhanced activation of NF-κB p65 and subsequent MMP9 production (Figure 5A). In the *Gpr81*^−/−^ mice’s IECs, we observed an increased degradation of IκBα, elevated MAPK activity (p38 and ERK), enhanced phospho-NF-κB p65 levels, and an increased production of MMP9 compared with the WT mice’s IECs (Figure 5A). Lactate functions as an endogenous agonist for GPR81 [16]. Utilizing lactate enabled us to selectively activate GPR81 and evaluate its consequential downstream effects within the context of this signaling pathway. Further treatment with lactate, which acts in a GPR81-dependent manner, inhibited NF-κB p65 activation and MMP9 production while preserving the expression of the TJ proteins Claudin-1, ZO-1, and Occludin in the IECs (Figure 5B). Based on these data (Figure 5C), lactate inhibits TNF-α-induced MMP-9 expression and thereby sustains TJ protein levels, protecting the integrity of intestinal barrier function by attenuating NF-κB activation via GPR81.

### 3.6. Lactate Attenuates Experimental Colitis in a GPR81-Dependent Manner

Considering the role of lactate/GPR81 signaling in modulating MMP-9 expression and the disruption of TJ proteins in IECs, we further evaluated the protective effect of this axis on intestinal barrier function in colitis. Lactate administration led to dose-dependent increases in body weight and colon lengths in the WT mice, whereas its effects were limited in the *Gpr81*^−/−^ mice after the DSS treatment (Figure 6A–C). A detailed histological analysis of colonic lesions showed that the lactate treatment significantly alleviated pathological alterations in the WT mice, including disruptions to the mucosal architecture, the infiltration of inflammatory cells, and areas of crypt loss. However, lactate intake did not significantly ameliorate DSS-induced colitis susceptibility in the *Gpr81*^−/−^ mice (Figure 6D). Furthermore, the protein expressions associated with the intestinal barrier—specifically MMP9, Claudin-1, ZO-1, and Occludin—were markedly increased in response to the lactate treatment in the WT mice but not in the *Gpr81*^−/−^ mice (Figure 6E). These findings suggest that lactate enhances intestinal barrier function in a GPR81-dependent manner.

## 4. Discussion

UC is a condition characterized by intestinal barrier dysfunction, which can be attributed to epithelial cell dysfunction or disruption caused by potent inflammatory mediators [5]. In this study, we examined the role of lactate/GPR81 signaling in intestinal epithelial barrier function and explored its potential mechanism. Our findings indicate that lactate/GPR81 signaling protects the integrity of the intestinal epithelial barrier and may alleviate colitis symptoms, possibly by inhibiting the activation of NF-κB/MMP9 pathways.

Recent clinical data suggest that UC is a progressive disease, characterized by changes in disease phenotype, adverse impacts on the bowel wall, increased risk of neoplasia, worsened colorectal function, and a higher likelihood of colectomy, hospitalizations, and extraintestinal comorbidities [2]. As a result, therapeutic goals for UC have shifted towards early aggressive intervention. Although anti-TNF-α biologics have significantly improved treatment outcomes for nearly two decades, a considerable number of patients do not respond to this therapy [42]. In our study, we found that the lactate/GPR81 axis could limit experimental colitis by inhibiting TNF-α/NF-κB/MMP9 signaling. This finding suggests that lactate could potentially serve as an alternative biological agent for alleviating early symptoms of colitis. Moreover, although data on the combination therapy of anti-TNF-α biological monotherapy are scarce, US guidelines recommend combining biologics to enhance the therapeutic response [3]. Therefore, combining lactate with an anti-TNF biologic may represent a potential strategy for relieving symptoms and reducing the risk of complications in patients with colitis. Dietary supplementation of lactate, including increased consumption of yogurt and lactic acid bacteria, may also prove to be efficacious in preventing and treating colitis. However, our study may only identify a potential colitis treatment strategy for patients with mild active colitis. Its effectiveness in patients with moderate or severe colitis requires further investigation.

GPR81, a G-coupled receptor, is highly expressed in adipose tissue and is also found in other tissues [43]. However, its functional role in the intestines remains unclear. Our findings suggest that GPR81 is frequently downregulated in colitis, indicating a potential role in gut inflammation. Previous studies have partially demonstrated the function of GPR81 in suppressing intestinal inflammation [21]. Our previous research has highlighted the critical role of GPR81 in resolving inflammation by inhibiting M1 macrophage polarization via A20 [44]. In this study, we further explored the protective function of GPR81 on intestinal barrier function in colitis. We found that the lactate/GPR81 axis enhances intestinal barrier function in colitis, aligning with earlier reports that laboratory-type symbionts or lactate supplements protect the gut against treatments such as radiation and chemotherapy by stimulating intestinal stem-cell-mediated epithelial growth through GPR81 [45]. Additionally, lactate, an endogenous ligand of GPR81, serves as an important precursor for the synthesis of SCFAs [14]. SCFAs play crucial roles in maintaining the epithelial layer’s integrity and facilitating tissue repair after mucosal damage. Previous research has demonstrated that butyrate ameliorates intestinal epithelial barrier dysfunction by activating GPR109A and inhibiting the AKT and NF-κB p65 signaling pathways [46]. In our study, we observed that lactate enhances the expression of Claudin-1, ZO-1, and Occludin via GPR81 and suppresses the NF-κB/MMP9 signaling pathways. Although our study primarily focuses on the protective role of Gpr81 in intestinal barrier function during colitis, the interaction between GPR81 and bacterial metabolites, particularly SCFAs, deserves further investigation.

Studies have identified that MMP9 plays a significant role in modulating inflammatory processes and tissue remodeling in colitis [47]. MMP9, produced by epithelial cells, can degrade collagen within the ECM, modulate cell–matrix interactions, and facilitate wound healing [48]. Moreover, MMP9 has been implicated in the degradation of TJ proteins, leading to increased intestinal permeability and exacerbating colitis [37,49,50]. Consistent with these findings, our data revealed that *Gpr81^−/−^* mice exhibit elevated levels of MMP9 expression and degradation of collagen and TJ proteins, resulting in profound epithelial barrier dysfunction and systemic inflammation in colitis. Furthermore, NF-κB, a key transcription factor, regulates most pro-inflammatory gene activations [51]. There is increasing evidence that MMP-9 can be induced by NF-κB [39,52]. Additionally, it has been shown that phosphorylated MAPK can activate the NF-κB pathway and elevate MMP9 expression. Studies have shown that propofol inhibits MMP-9 production in endothelial cells by inhibiting the Ca^2+^/CAMKII/ERK/NF-κB signaling pathway [53]. It has also been reported that MMP9-mediated proteolytic degradation involving p38 may reduce Occludin levels [41]. Consistently, our investigation found that TNF-α induces the phosphorylation of NF-κB and MMP9 expression by promoting IκBα degradation and activating ERK and p38. However, the lactate/GPR81 axis attenuates this effect.

In addition to the tight-junction proteins and metabolites we have explored, it is essential to acknowledge the crucial role of the intestinal mucus layer, goblet cells, and immune cells in maintaining the integrity of the intestinal barrier. It is well known that an intact mucosal barrier is required to keep bacteria at a distance from the epithelium, thereby emphasizing the significance of the mucosal components in human health and disease [54]. The intestinal mucus layer, primarily secreted by goblet cells, contains antimicrobial peptides and immunoglobulins, which contribute to mucosal defense and homeostasis. Goblet cells play a vital role in replenishing and maintaining the mucus layer through the production of mucins, thus strengthening the barrier [55]. Furthermore, dietary intake profoundly influences the composition of the gut microbiome, which in turn affects the integrity of the intestinal mucus layer, impacting the homeostasis of the intestinal barrier and overall gut health [56]. Immune cells within the intestinal mucosa, such as intraepithelial lymphocytes, dendritic cells, and macrophages, actively participate in immune surveillance and regulate barrier function. These immune cells coordinate inflammatory responses, regulate epithelial cell turnover, and promote tissue repair in response to injury [57]. Although our current study did not investigate these components, we recognize this as a limitation and recommend future studies examine the comprehensive interaction of these factors with the intestinal epithelium to gain a more holistic understanding of barrier integrity in the context of UC. Another limitation is the lack of IEC-specific knockout mice. Incorporating this animal model in future studies could provide greater clarity on the specific role of GPR81 in intestinal epithelial barrier function. Furthermore, while we have demonstrated the significant impact of the lactate/GPR81 axis on TNF-α-induced NF-κB/MMP9 signaling, further investigation is required to fully comprehend the complex molecular interplay between GPR81 and NF-κB activation. Lastly, future research directions should involve expanding sample collection, including paired samples from UC patients and samples from other intestinal diseases, to further understand the role of GPR81 in intestinal inflammation and its disease specificity.

## 5. Conclusions

In conclusion, our study highlights the role of GPR81 as a crucial homeostatic regulator that mitigates oxidative stress, attenuates colonic inflammation, and preserves the intestine epithelial barrier function in DSS-treated mice. The lactate/GPR81 axis plays a vital role in maintaining tight junctions and barrier integrity among intestinal epithelial cells by suppressing the hyperactivation of the NF-κB/MMP9 signaling pathway. Additionally, our findings suggest that lactate supplementation can effectively protect against DSS-induced colitis in a GPR81-dependent manner. These findings suggest that lactate, acting as the endogenous ligand for GPR81, may be explored as a potential biological agent in the treatment of UC.

## Figures and Tables

**Figure 1 nutrients-16-00582-f001:**
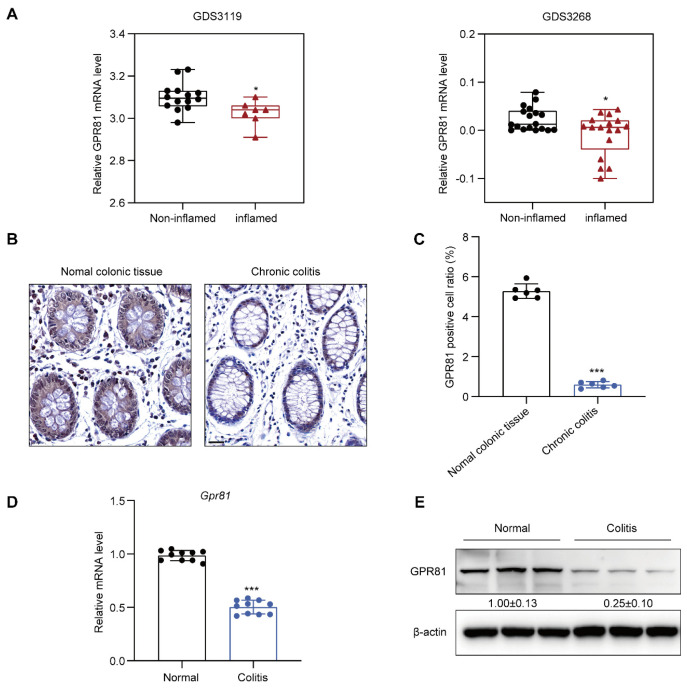
Downregulation of GPR81 expression in the intestinal mucosal tissue of colitis patients. (**A**) The mRNA level of GPR81 in inflamed and non-inflamed colon tissues from GEO datasets. (**B**) Immunohistochemical staining images depicting GPR81 expression on a colon tissue microarray (highlighted in yellow), with a scale of 20 μm. (**C**) Quantitative analysis of positive cells in IHC staining (*n* = 6). (**D**) qPCR analysis of *Gpr81* mRNA expression in mouse colon mucosal tissue (*n* = 10). (**E**) Protein expression of GPR81 in mouse colon mucosal tissue. * *p* < 0.05, and *** *p* < 0.005.

**Figure 2 nutrients-16-00582-f002:**
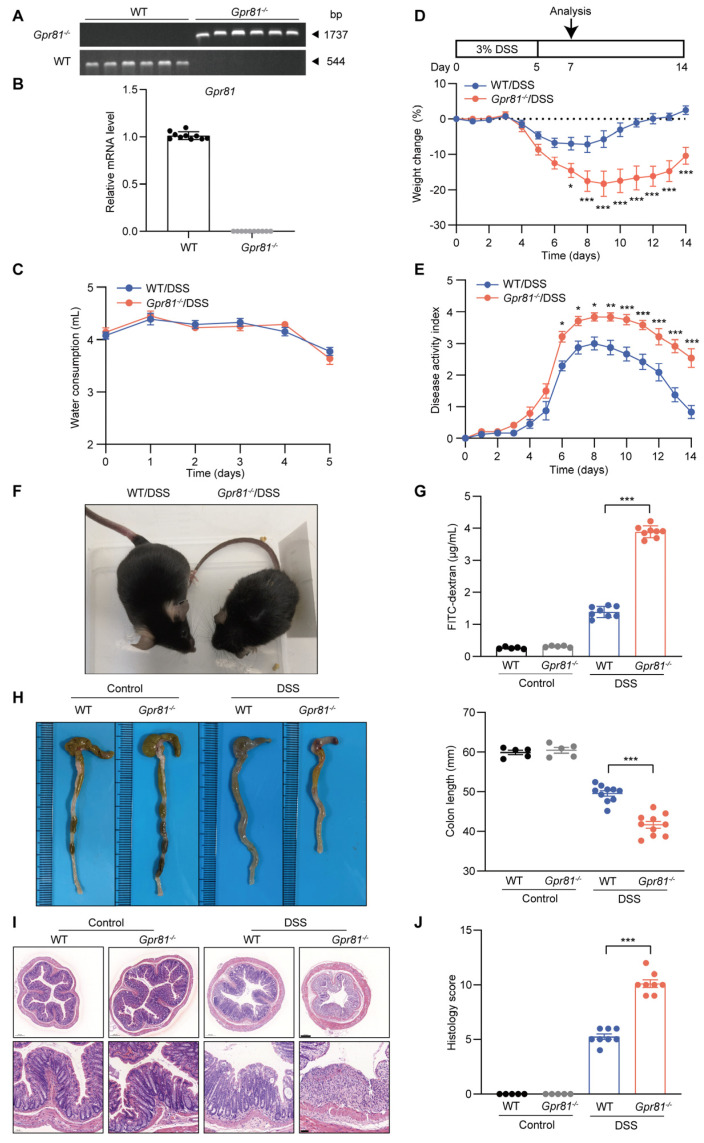
*Gpr81*-deficient mice exhibit high susceptibility to DSS-induced colitis. (**A**)Agarose gel electrophoresis of PCR products from WT mice and *Gpr81^−/−^* mice (*n* = 6). (**B**) qPCR analysis of *Gpr81* mRNA expression in mouse colon mucosal tissue (*n* = 10). (**C**) Water consumption during DSS treatment. (**D**) Experimental design and relative weight changes in WT or *Gpr81^−/−^* mice (*n* = 8). (**E**) Disease activity indices (DAIs), comprising averages of body weight loss, fecal consistency, and fecal blood test scores (*n* = 8). (**F**) Visual images depicting the condition of mice. (**G**) Serum FITC–dextran concentration (*n* = 5 or 8). (**H**) Representative images and lengths of colons (*n* = 5 or 8). (**I**) H&E staining, with scale bars of 200 μm (bottom) and 50 μm (upper). (**J**) Colitis severity scores (*n* = 5 or 8). * *p* < 0.05; ** *p* < 0.01; *** *p* < 0.005.

**Figure 3 nutrients-16-00582-f003:**
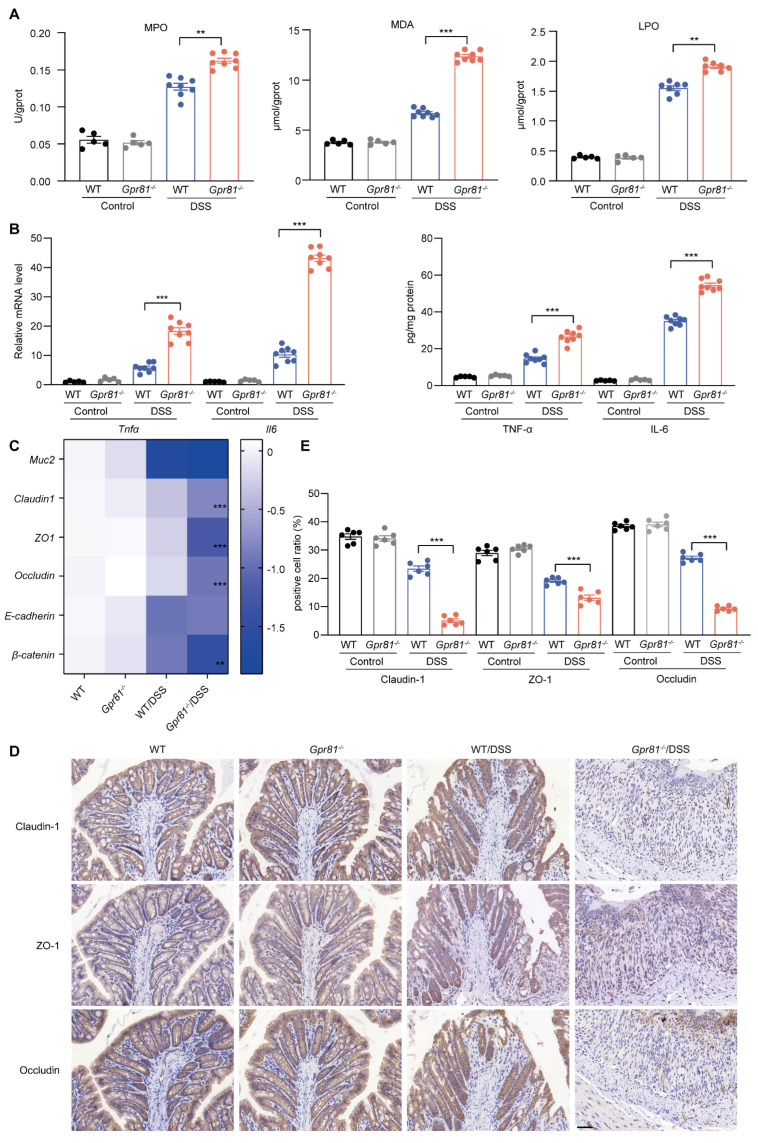
*Gpr81* deficiency impairs the intestinal epithelial barrier function in colitis. (**A**) The activity of MPO, as well as the concentrations of MDA and LPO, in colon tissues (*n* = 5 or 8). (**B**) Relative mRNA and protein expression levels of cytokines in colon tissues (*n* = 5 or 8). (**C**) Expression of genes related to the intestinal epithelial barrier function (*n* = 5 or 8). (**D**) Immunohistochemical staining of Claudin-1-, ZO-1-, and Occludin-positive cells in colon tissues, with a scale of 50 μm (*n* = 5). (**E**) Quantitative analysis of positive cells in IHC staining (*n* = 6). ** *p* < 0.01; *** *p* < 0.005.

**Figure 4 nutrients-16-00582-f004:**
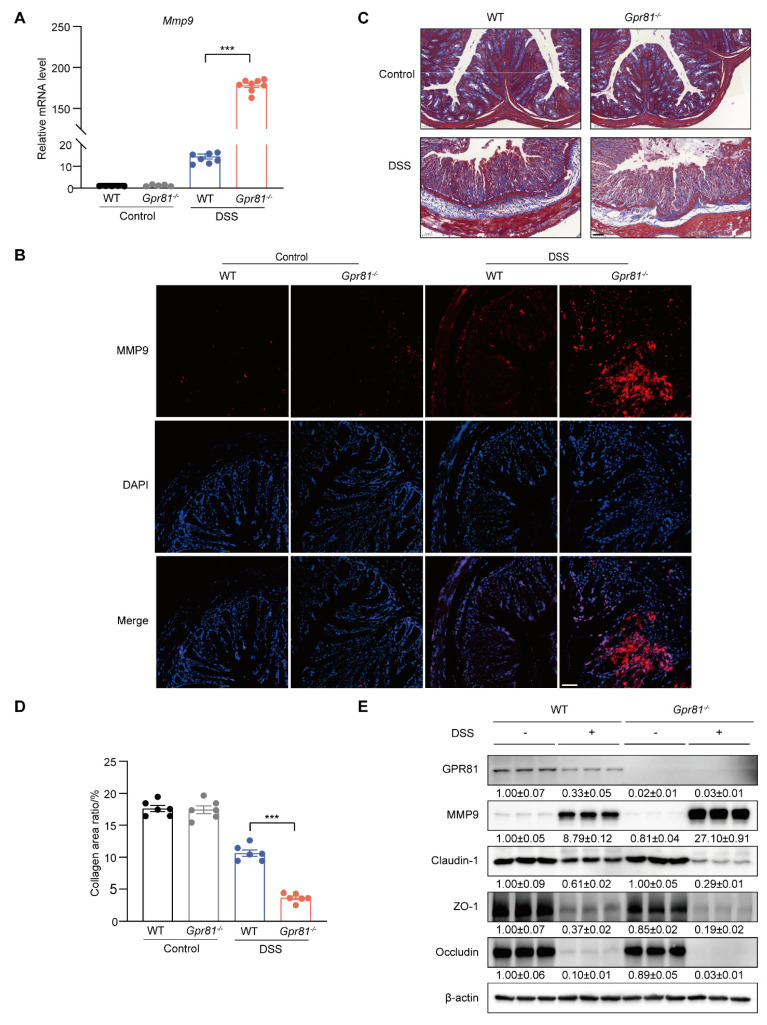
*Gpr81* deficiency promotes MMP9 production in colitis. (**A**) Relative mRNA level of *Mmp9* (*n* = 5 or 8). (**B**) IF analysis of MMP9 (red) counterstained with DAPI in the colon, with a scale of 50 μm (*n* = 5). (**C**) Representative images of Masson’s trichrome staining for collagen (blue) in colons from WT and *Gpr81^−/−^* mice (*n* = 5). (**D**) Quantitative analysis of collagen using Masson’s trichrome staining (*n* = 6). (**E**) Protein expression levels of GPR81, MMP9, Claudin-1, ZO-1, and Occludin in colon homogenate. *** *p* < 0.005.

**Figure 5 nutrients-16-00582-f005:**
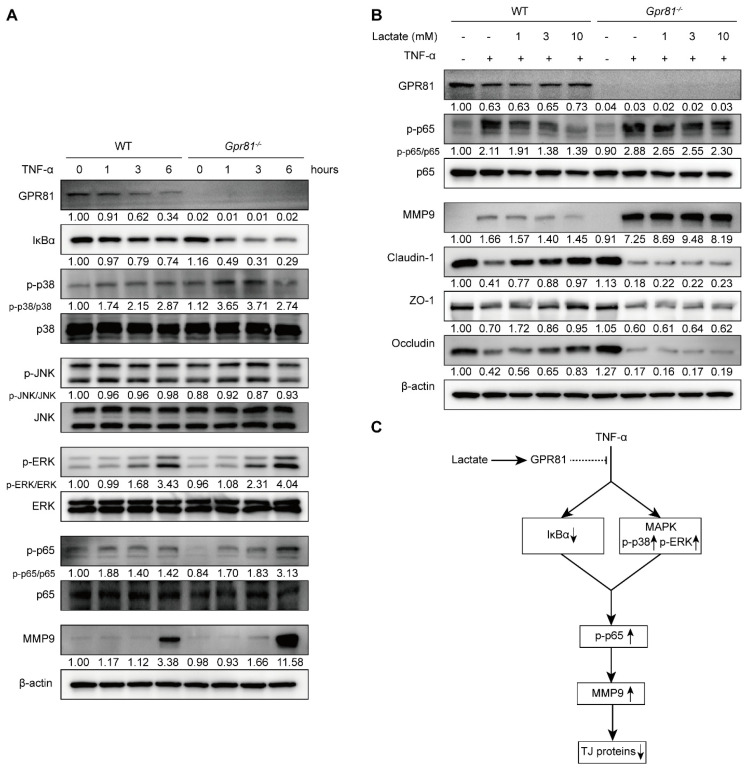
Lactate/GPR81 signaling modulates TNF-α-induced MMP9 expression in IECs via the NF-κB pathway. (**A**) The influence of *Gpr81* deficiency on NF-κB pathway-related protein expression and MMP9 in colonic IECs after TNF-α treatment. (**B**) The effect of lactate on the expression of p-p65, MMP9, Claudin-1, ZO-1, and Occludin in colonic IECs treated with TNF-α treatment for 6 h. (**C**) A schematic representation illustrating the modulation of TNF-α-induced MMP-9 expression in IECs by lactate/GPR81 signaling. Down arrow indicates downregulation, and up arrow indicates upregulation.

**Figure 6 nutrients-16-00582-f006:**
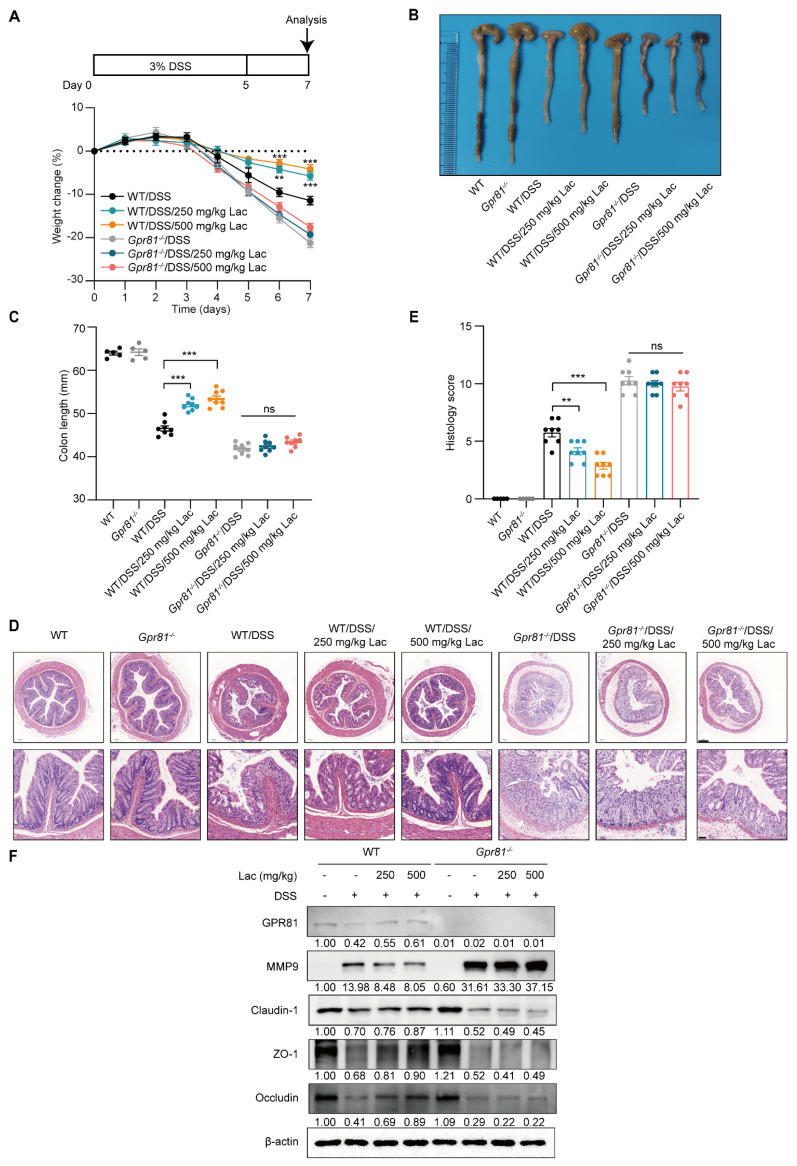
Lactate attenuates colitis in a GPR81-dependent manner in mice. (**A**) Experimental design and changes in body weight (*n* = 8). (**B**) Representative images of colons. (**C**) Measurements of colon length (*n* = 5 or 8). (**D**) H&E staining, with scale bars of 200 μm (bottom) and 50 μm (upper). (**E**) Colitis severity scores (*n* = 5 or 8). (**F**) Expression levels of GPR81, MMP9, Claudin-1, ZO-1, and Occludin in colon homogenates. ** *p* < 0.01; *** *p* < 0.005. “ns” denotes non-significant differences.

**Table 1 nutrients-16-00582-t001:** Primer sequences.

Gene	Primer Sequences
*Gpr81*-WT-F	TAGAGCAGGGACCCGACTTC
*Gpr81*-WT-R	GATCGAGCCCTGGAGATGAC
*Gpr81*^−/−^-F	CGCAAGCGTGCATATTCTGG
*Gpr81*^−/−^-R	TGGGTCCTCATGGTCATGTG
*β-actin*-F	GGCTGTATTCCCCTCCATCG
*β-actin*-R	CCAGTTGGTAACAATGCCATGT
*Gpr81*-F	GCTTACCCCTTCGGACAGAC
*Gpr81-R*	ATGCTCCCGGCCCTATTCA
*Tnfα*-F	CAGGCGGTGCCTATGTCTC
*Tnfα*-R	CGATCACCCCGAAGTTCAGTAG
*Il6*-F	TTGGTCCTTAGCCACTCCTCC
*Il6*-R	TAGTCCTTCCTACCCCAATTTCC
*Muc2*-F	ATGCCCACCTCCTCAAAGAC
*Muc2*-R	GTAGTTTCCGTTGGAACAGTGAA
*Claudin1*-F	GGGGACAACATCGTGACCG
*Claudin1*-R	AGGAGTCGAAGACTTTGCACT
*Zo1*-F	GCCGCTAAGAGCACAGCAA
*Zo1*-R	GCCCTCCTTTTAACACATCAGA
*Occludin*-F	TTGAAAGTCCACCTCCTTACAGA
*Occludin*-R	CCGGATAAAAAGAGTACGCTGG
*β-catenin*-F	ATGGAGCCGGACAGAAAAGC
*β-catenin*-R	CTTGCCACTCAGGGAAGGA
*E-cadherin*-F	CAGGTCTCCTCATGGCTTTGC
*E-cadherin*-R	CTTCCGAAAAGAAGGCTGTCC
*Mmp9*-F	GCCCTGGAACTCACACGACA
*Mmp9*-R	TTGGAAACTCACACGCCAGAAG

## Data Availability

The data presented in this study are available on request from the corresponding author. The data are not publicly available due to privacy.

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
