# Peer review of "Lactate Protects Intestinal Epithelial Barrier Function from Dextran Sulfate Sodium-Induced Damage by GPR81 Signaling"

_nutrients, 2024, doi:10.3390/nu16050582_

Round 1

Reviewer 1 Report

Comments and Suggestions for Authors

In this study, the authors aimed at exploring the potential role of GPR81 in regulating epithelial barrier integrity and preventing leaky gut in colitis. The story is interesting, but the manuscript needs a major revision before considering for a publication. 

here my major comments

lines 68-70. It would be reductive to define lactate that way. Several studies reported lactate not only as a substrate providing energy to support cell growth, but it acts as signal molecule affecting and regulating biochemical and biological function. For example, lactate is involved in immune cell fate and function. (doi: 10.3389/fendo.2022.901495x; doi: 10.1038/s41392-022-01206-5; doi.org/10.3389/fphys.2021.688485; doi.org/10.3389/fimmu.2021.808799). therefore, the sentence need to be improved.

lines 102-112. Please, indicate the route of administration of the lactate and  justify the choice of only male mice.

lines 114-120. There are no indication related to antibody concentration.

lines 129-131. Please, indicate the code and brand of the products.

lines 142-143. Please, specify the transfection reagent.

lines 162. Please, indicate antibody concentration.

Figure legends. Please, indicate the sample number used in each experiment.

lines 179. Have the authors quantified the GPR81 levels in the human colon tissue microarray from normal and UC samples? Could you please include the quantification in Figure 1B?

lines 188. The yellow highlight is not visible in Figure 1B.

line 181. Could you please provide the western blot quantification for GPR81 in Figure 1D?

Have you evaluated on which cells GPR81 is expressed (epithelial cells, fibroblasts and/or epithelial cells)? Have you performed staining of co-localization?

line 204. Please, indicate in the text when permeability assay and colon collection were performed. During the acute or the recovery phase of colitis? Do you evaluate the mucosal state during the acute or recovery phase?

line 208. Please, combine figure 2E and F because are redudant.

line 213. If increased permeability and severity of colitis were observed during the regeneration phase, please add in the final sentence that the protective function of GPR81 is related to the recovery phase of colitis.

line 235. Please, provide quantification of IHC for Claudin-1, ZO-1 and Occludin in Figure 3D and increase the quality and magnifiation of the figure

line 253. The quality and magnifiation of the figure Figure 4B should be improved

lines 255-259.  quantification of collagen deposition and western blot in Figure 4C and 4D, respectively should be included

lines 268-281. Please, provide western blot quantification in Figure 5A, 5B, and 5C and normalized the phosphorylated proteins on the corresponding total proteins.

line 272. Please, justify why TNF-a is used. To mimic an inflammatory stimulus in healthy IECs?

line 274. Specify if the proteins used for western blot were extracted from IECs isolated from Gpr81-/- mice or from IECs isolated from WT mice and subjected to silencing with GPR81 siRNA. From the text seems that the protein analysis was performed on IECs derived from GPR81-/- mice, but in the Figure 5B and C appeared siGPR81. Please, clarify.

line 276. Justify in more detail why lactate is used.

line 287. Specify the timing of the stimulation in the Figure legend 5C. it is not clear if it was performed  6h after TNF-a stimulation

Have you inhibited the ERK and NF-kB pathway in order to demonstrate their involvement in the lactate/GPR81 axis?

lines 295-298. Please, provide the histological score in Figure 6D.

line 300. Please, provide the western blot quantification in Figure 6E.

Is lactate able to restore oxidative stress and the inflammatory response induced by DSS in WT mice?

Comments on the Quality of English Language

The manuscript needs a thorough revision to improve methods, figure legends, figures and results. 

Reviewer 2 Report

Comments and Suggestions for Authors

In the following manuscript Li et al., have investigated in a DSS-induced colitis model the role of GPR81 in colitis progression and the impact of lactate/GPR81 signaling on the intestinal epithelial barrier function. The study is well conducted, and the manuscript is easy to follow. However, there are important points that must be addressed by the authors. Please see my comments below:

Major comments

Line 47: This part needs to be further extended since the authors have not included information linked to the importance of the intestinal mucus layers and the cells (i.e., goblet cells) devoted to its production in the context of inflammatory bowel diseases. It is well known that an intact mucosal barrier is required to keep bacteria at a distance from the epithelium, thereby underlying the importance of the slime partner in human health and disease. Two recent reviews in the field of diet-microbiota-host interactions summarize the importance of diets, glycans, and the intestinal mucus layer in human health and disease (please see Suriano et al., Frontiers in Immunology, 2022, and Luis & Hansson Cell Host & Microbiome, 2023). Please refer to these two reviews since the authors have summarized most of the recent studies in this field.

I assume and according to the full story of the manuscript, that the Gpr81 gene deletion occurs in a specific tissue (i.e., intestine). If this is the case, the authors should provide a result confirming the full deletion of the concerned gene in the targeted tissue. Moreover, is the deletion specific to the proximal or the distal part of the gut? Please further elaborate in this regard and provide additional results.

Can deletion of Gpr81 be associated with a lower/higher production of bacterial metabolites (e.g., SCFAs)? Do the authors have those additional results?

General remark for the discussion and conclusion, please keep in mind that an intact intestinal epithelial barrier is not only associated with tight junction proteins, metabolites, etc., but the intestinal mucus layer, goblet cells, and immune cells also play a pivotal role. Please further elaborate on this concept in the discussion and provide a limitation of the following study since the authors have not looked at it.

Lines 388-390: Quite speculative, please rephrase it.

Minor comments

Lines 309-316 For ulcerative colitis, please use the abbreviation UC for the full text.

Line 335: Please replace Gi with G

Reviewer 3 Report

Comments and Suggestions for Authors

The authors aimed to investigate the mucosal GPR81 signaling activated by lactate in the protection of DSS colitis. Overall, the stud looks interesting with some mechanistic experiments. Please see my comments shown below.

1. What do the authors want to say "normal tissue" in Figure 1? (endoscopically-non-inflamed tissues or healthy controls? ) The detailed information of Figure 1B is missing (describe the patient's information). 

2. Please show the expression levels of GPR81 in the inflamed and non-inflamed tissues from the same individuals of UC patients. It is also needed to show the GPR81 expression levels in other intestinal diseases including Crohn's disease if the abnormal GPR81 levels is specific to UC. 

3. Please show or describe the consumption of DSS in each group.

4. To investigate mucosal permeability, a FITC dextran assay would be helpful.

5. Please show the quantitative data associated with western blots as well.

Round 2

Reviewer 1 Report

Comments and Suggestions for Authors

No additional comments

Comments on the Quality of English Language

the english should be revised for minor errors

Author Response

Dear Reviewer,

Thank you for your constructive feedback. We appreciate the opportunity to improve the clarity and precision of our writing.

In response to your comment, we have meticulously reviewed the entire manuscript and performed a comprehensive language revision to address and correct all minor English errors. This includes grammatical adjustments, corrections to specialized vocabulary, and enhancements to sentence structure for improved readability and academic rigor.

Thank you once again for your valuable feedback. 

Sincerely,

Haitao Li

Reviewer 2 Report

Comments and Suggestions for Authors

I have no additional comments to be addressed to the authors.

One minor comment only:

Lines 400-401: Please use the term intestinal mucus layer, and not mucosal. The latter is not appropriate. Please carefully check that this is applied for the whole text when referring to the intestinal mucus layer. 

Author Response

Dear Reviewer,

Thank you very much for your insightful observation regarding the use of the term "intestinal mucus layer" as opposed to "mucosal." 

We have corrected it according to your opinion. Additionally, we have carefully reviewed the entire manuscript and this adjustment has been applied throughout the manuscript to maintain accuracy and appropriateness of the terminology.

We are grateful once again for your attentive review and helpful suggestions.

Sincerely,

Haitao Li